# Multimodal Generative Learning on the MIMIC-CXR Database

**Hendrik Klug**[1]                                                    KLUGH@ETHZ.CH

[1] *Department of Electrical Engineering, ETH Zürich*

**Thomas M. Sutter**[2]                                      THOMAS.SUTTER@INF.ETHZ.CH

**Julia E. Vogt**[2]                                              JULIA.VOGT@INF.ETHZ.CH

[2] *Department of Computer Science, ETH Zürich*

## Abstract

Machine Learning has become more and more popular in the medical domain over the past years. While supervised machine learning has already been applied successfully, the vast amount of unlabelled data offers new opportunities for un- and self-supervised learning methods. Especially with regard to the multimodal nature of most clinical data, the labelling of multiple data types becomes quickly infeasible in the medical domain. However, to the best of our knowledge, multimodal, unsupervised and generative methods have been tested extensively on toy-datasets only but have never been applied to real-world medical data, for direct applications such as disease classification and image generation. In this article, we demonstrate that this class of methods provides promising results on medical data while highlighting that the task is extremely challenging and that there is space for substantial improvements.

**Keywords:** Multimodal Learning, Generative Learning, VAE

## 1. Introduction

Clinical data is usually gathered in many modalities, such as images, text reports or electronic health records. A generative model that can leverage all available modalities could be useful for important tasks such as generating text reports when images of a patient are given, generating an image of another angle from an input image or improved data classification given all modalities. Especially in the medical domain, obtaining annotations of sufficient training examples for the training of a deep learning model is expensive, since it requires manual expert input. If many modalities are available, this quickly becomes infeasible. A self-supervised generative model is able to benefit from the large pool of unlabelled data since it is able to learn from data without the need for labels.

The Variational Autoencoder (Kingma and Welling, 2014, VAE) in particular is a popular generative model, which consists of an encoder that maps the input to a learned latent distribution from which the decoder part samples to reconstruct the input. In contrast to toy datasets used in previous work on multimodal generative learning, the different pathologies that represent the classes in medical data are defined by very subtle features that are difficult to recognise even for human experts. This makes the task of learning a latent representation of the data, while preserving the separability of the classes, extremely challenging.

In this work, we apply the generalised multimodal ELBO (Sutter et al., 2021) to train a self-supervised, generative model on clinical data from the MIMIC-CXR Database (Johnson et al., 2019) containing chest-X rays and related text reports. We provide a first insight into the difficulties and opportunities that come with medical data.

## 2. Methods & Results

Assuming the data consists of $N$ i.i.d. samples $\{\mathbb{X}^{(i)}\}_{i=1}^{N}$, each of which is a set of M modalities $\mathbb{X}^{(i)} = \{\mathbf{x}_j^{(i)}\}_{j=1}^{M}$, the joint posterior distribution is calculated in two steps. In a first step, a distribution is calculated for each subset of the powerset $\mathcal{P}(\mathbb{X})$ using a Product of Experts (Wu and Goodman, 2018, PoE). In a second step these subsets are merged using a Mixture of Experts (Shi et al., 2019, MoE) into a joint posterior distribution. For a more detailed explanation, we refer to the original paper (Sutter et al., 2021).

The MIMIC-CXR Database (Johnson et al., 2019) provides multiple class-labels for every sample where each class corresponds to one of 13 pathologies. For the evaluation of our method, we created a label "Finding", indicating if the sample is labeled with any of the 13 pathologies.

**Evaluation of the latent representation.** We evaluate the ability of a linear classifier to classify subsets of different modalities according to the label "Finding". The evaluation with respect to every subset of modalities demonstrates the model's ability to infer meaningful representations in case of missing data types. Table 1 shows the performance for all subsets of the powerset $\mathcal{P}(\mathbb{X})$. The highest score is achieved if all three modalities are present. This demonstrates that the information of all modalities is successfully merged in the joint posterior distribution.

Table 1: **Classification results of the linear classifiers.** We evaluate the ability of the classifiers to classify the learned representations of all subsets of modalities correctly. The mean average precision over the test set is reported (F: frontal image, L: lateral image, T: text report).

| MODEL | F | L | T | L,F | F,T | L,T | L,F,T |
|---|---|---|---|---|---|---|---|
| MoPoE | 0.467 | 0.460 | 0.473 | 0.476 | 0.493 | 0.475 | **0.494** |
| Random | | | 0.235 | | | | |

**Generation Quality.** Figures 1(a) and 1(b) compare the generation quality of the model, with and without the F modality as input. Adding the F modality as input brings a significant improvement to the quality of the generated images. The generated samples from Figure 1(b) are less blurry and smaller details are recognisable.

## 3. Conclusion

In this work, we apply a method for multimodal generative learning on clinical data and evaluate its performance on direct applications in the medical field. While our experiments show promising results, they indicate that the task is extremely challenging with significant scope for improvement. In particular, features in medical scans that are characteristic of most pathologies are often small details that get lost in the blurriness of the generated samples. However, the latent representation that the MoPoE learns when learning to reproduce the data is still meaningful in the sense that it can be separated into the different classes the data belong to. We argue that our method is a successful first step into creating an unsupervised method that will find applications in the medical domain such as classification

of diseases, generating text reports from medical data and generating scans from multiple angles.

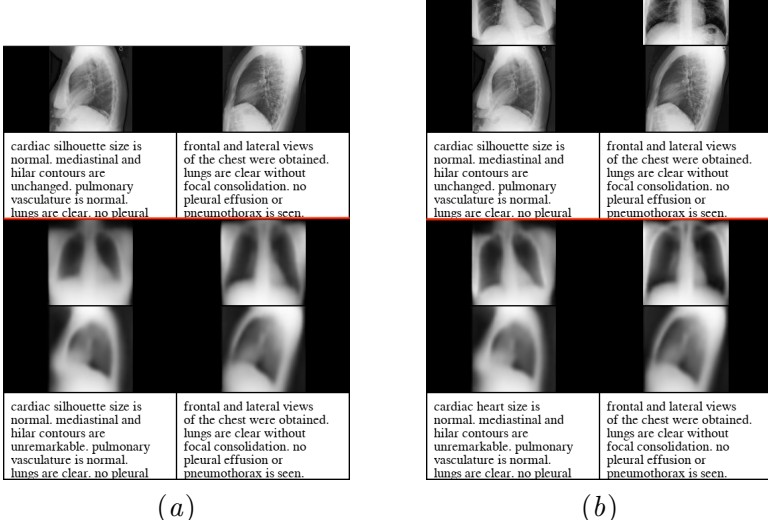

Figure 1: **Generated samples.** On the left, the L and T modality are given to the model as input. On the right, all modalities (F, L and T) are given as input. The samples above the red line are the input samples and those below are generated by the model.

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
