# OpenReview forum: "Multimodal Generative Learning on the MIMIC-CXR Database"
_MIDL.io/2021/Conference/Short — MIDL 2021 Poster_

### Official Review · Reviewer_3TJm · 2021-04-29

**Confidence:** 4
**Final Rating:** 3

**Summary:**

The authors fit a multi-modal generative model, which they themselves have developed (ICLR 2021), to MIMIC-CXR multi-modal data: CXRs and radiological reports. Their validation classifies the learned latent representation into different findings, and qualitatively shows generated samples. This could very well be one of the first applications of a deep generative model to multimodal - that is, text and images - medical data.

**Strengths:**

It is a well written paper that flows well. I think the community will find it interesting, as its application has not been well explored (yet) with deep generative models. The paper also demonstrates a theoretical framework for doing so, that many readers probably will find interesting, too.

**Weaknesses:**

My main concern with this paper is the low precision from the latent space classification experiment. In the Conclusion section, the authors state ‘However, the latent representation that the MoPoE learns when learning to reproduce the data is still meaningful in the sense that it can be separated into the different classes the data belong to.’ However, a precision below 0.5 effectively means the classification predicts more false positives than true positives; therefore I am not sure their claim is justified, meaning how useful is the latent representation their model encodes? Also, I am not sure that picking frontal and lateral slices from the CXR images qualifies as different modalities? (if I understood the meaning of the F and L images - it is not explained in the paper). Why not simply use two modalities, 3D CXR and text reports?

**Deanonymize Review:**

no

**Detailed Comments:**

* Your ICLR citation shows the braced brackets (for converting to caps).
* ‘multimodal unsupervised methods have been tested extensively on toy-datasets only but have never been applied to real-world medical data’, this statement in the abstract is incorrect, unsupervised ML models have been used for years at this task, see for example:
Groves, Adrian R., et al. "Linked independent component analysis for multimodal data fusion." Neuroimage 54.3 (2011): 2198-2217.
Maybe you mean deep generative models?
* It would be interesting to get details on how you encode the text.
* I am a bit confused as to what you mean by conditionally generated samples (Fig. 1)? I suspect that it simply means reconstructing samples with your learned VAE, i.e., passing the top three rows of (a) or (b) through a forward pass? Or does it mean conditional as in a CVAE?

**Justification Of The Rating:**

The authors should be commended for taking on such a difficult problem: to model a joint distribution of data as different and heterogeneous as CXRs and radiological reports. Their application is an interesting field of research, with plenty of potential, and they are applying a novel method that the community might find interesting. I think this in probably warrants publication at MIDL, although the classification results are low.

**Paper Type:**

validation/application paper

**Special Issue:**

no

---

### Meta-Review · Area_Chair_73YW · 2021-05-09

**Recommendation:** Accept (Poster)
**Confidence:** 4

**Metareview:**

The paper addresses a challenging yet important issue of multimodal learning (images and reports) with deep generative models. The reviewer is fairly convinced by the proposed method (based on ICLR paper) and the potential for the application. However, the results are currently of somewhat performance. The work will likely stimulate fruitful discussions and can therefore be accepted as short paper.

---

### Decision · Program_Chairs · 2021-05-11

Accept (Poster)